# Inter-Comparison of UT1-UTC from 24-Hour, Intensives, and VGOS Sessions during CONT17

**DOI:** 10.3390/s22072740

**Published:** 2022-04-02

**Authors:** Shrishail Raut, Robert Heinkelmann, Sadegh Modiri, Santiago Belda, Kyriakos Balidakis, Harald Schuh

**Affiliations:** 1GFZ German Research Centre for Geosciences, 14473 Potsdam, Germany; rob@gfz-potsdam.de (R.H.); kyriakos.balidakis@gfz-potsdam.de (K.B.); schuh@gfz-potsdam.de (H.S.); 2Chair of Satellite Geodesy, Institute of Geodesy and Geoinformation Science, Technische Universität Berlin, 10623 Berlin, Germany; 3Department Geodesy, Federal Agency for Cartography and Geodesy (BKG), 60322 Frankfurt am Main, Germany; sadegh.modiri@bkg.bund.de; 4UAVAC, Applied Mathematics Department, University of Alicante, 03690 Alicante, Spain; santiago.belda@uv.es

**Keywords:** UT1-UTC, VLBI, CONT17, VGOS, Intensive session

## Abstract

This work focuses on the assessment of UT1-UTC estimates from various types of sessions during the CONT17 campaign. We chose the CONT17 campaign as it provides 15 days of continuous, high-quality VLBI data from two legacy networks (S/X band), i.e., Legacy-1 (IVS) and Legacy-2 (VLBA) (having different network geometry and are non-overlapping), two types of Intensive sessions, i.e., IVS and Russian Intensives, and five days of new-generation, broadband VGOS sessions. This work also investigates different approaches to optimally compare dUT1 from Intensives with respect to the 24 h sessions given the different parameterization adopted for analyzing Intensives and different session lengths. One approach includes the estimation of dUT1 from pseudo Intensives, which are created from the 24 h sessions having their epochs synchronized with respect to the Intensive sessions. Besides, we assessed the quality of the dUT1 estimated from VGOS sessions at daily and sub-daily resolution. The study suggests that a different approach should be adopted when comparing the dUT1 from the Intensives, i.e., comparison of dUT1 value at the mean epoch of an Intensive session. The initial results regarding the VGOS sessions show that the dUT1 estimated from VGOS shows good agreement with the legacy network despite featuring fewer observations and stations. In the case of sub-daily dUT1 from VGOS sessions, we found that estimating dUT1 with 6 h resolution is superior to other sub-daily resolutions. Moreover, we introduced a new concept of sub-daily dUT1-tie to improve the estimation of dUT1 from the Intensive sessions. We observed an improvement of up to 20% with respect to the dUT1 from the 24 h sessions.

## 1. Introduction

Very long baseline interferometry (VLBI) is a microwave-based space geodetic technique that measures the difference in arrival time of signals from extra-galactic radio sources (e.g., the quasars) received simultaneously at two or more radio telescopes [1]. VLBI is the only high-precision space geodetic technique that can provide the full set of the Earth orientation parameters (EOP). The EOP provide the link between the international terrestrial reference system (ITRS) and the geocentric celestial reference system (GCRS). The EOP consist of five angles, namely, polar motion (PM), given by $x_{p}$ and $y_{p}$, Earth rotation angle (ERA), that is linearly proportional to UT1 (dUT1 or UT1-UTC), and celestial pole offsets (CPO) (dX,dY) [2]. In this study, we investigate UT1-UTC from VLBI sessions. VLBI is the only space geodetic technique that can determine the ERA. The relationship between UT1 and ERA is defined by Equation (Equation 1) [3]:(1)$$\theta \left(T_{u}\right)=2\pi \cdot \left(0.7790572732640+1.00273781191135448\cdot T_{u}\right)$$
where $T_{u}=\left(Julian\;\mathrm{UT}1\;date-2451545.0\right)$, and UT1=UTC+(UT1−UTC). Other space geodetic techniques, such as the Global Navigation Satellite Systems (GNSS), require VLBI-determined UT1-UTC as they can only estimate the (negative) time derivative of UT1-UTC, the length of day (LOD). The quantity dUT1 is required for the precise transformation of satellite positions from Earth-centered inertial (ECI) to Earth-centered Earth-fixed (ECEF) frame. Hence, for satellite-based space geodetic techniques, this quantity has to be provided externally with the best available resolution and accuracy.

The International VLBI Service for Geodesy and Astrometry (IVS) established the so-called Intensive session program, which has been carried out since 1984 (https://ivscc.gsfc.nasa.gov/index.html, accessed on 1 March 2022). The sole objective of Intensive sessions is to determine UT1-UTC in a timely manner, and they consist mainly of one VLBI baseline. Such sessions typically last for approximately one hour. The baselines are typically chosen, providing a long east–west extension, as dUT1 is highly sensitive to it [4]. We can understand this from the following equation, which shows the partial derivative of the VLBI group delay observable, τ, with respect to UT1 [5]:(2)$$\frac{\partial \tau }{\partial \mathrm{UT}1}=\frac{1}{c}\left(k_{x}\left[b_{x}\sin\theta +b_{y}\cos\theta \right]+k_{y}\left[-b_{x}\cos\theta +b_{y}\sin\theta \right]\right)\frac{\partial \theta }{\partial \mathrm{UT}1}$$
where *c* is the speed of light, $b_{x}$ and $b_{y}$ are the baseline components in *x* and *y* direction, respectively, in the Terrestrial Intermediate Reference System (i.e., the terrestrial system at observing epoch with polar motion considered), $k_{x}$ and $k_{y}$ are the radio source components in *x* and *y* direction, respectively, in the Celestial Intermediate Reference System (i.e., the celestial system at observing epoch with precession/nutations considered), and $\frac{\partial \theta }{\partial \mathrm{UT}1}=1.00273781191135448$ [2]. To obtain a high sensitivity for dUT1, a large $\frac{\partial \tau }{\partial \mathrm{UT}1}$ is required, as seen from Equation (Equation 2). If the differences in *x* and *y* station coordinates $b_{x}=\left(x_{2}-x_{1}\right)$ and $b_{y}=\left(y_{2}-y_{1}\right)$, respectively, are large, the sensitivity will be large as well. Hence, to have accurate dUT1 values, $b_{xy}=\sqrt{\left(b_{x}^{2}+b_{y}^{2}\right)}$ should be large. Besides, Ref. [6] investigated the dependence of longitudinal span on dUT1 results and concluded that, empirically, the UT1 uncertainty is inversely proportional to the longitudinal span.

It is of vital importance that the dUT1 estimated from Intensives should have the best accuracy possible. Recently, several studies have been conducted regarding possible improvements of dUT1 results ([7,8]). Gipson and Baver [7] observed an improvement in UT1 estimates from IVS-INT01 sessions by adapting a new source selection strategy. Nilsson et al. [8] concluded that dUT1 estimates are significantly affected by the tropospheric delays and tropospheric gradients. The study recommends applying a priori tropospheric gradients computed from a numerical weather prediction model or GNSS instead of applying empirical models such as the APG model [9]. Besides, errors in the PM and nutation models can affect the accuracy of dUT1 estimates [10]. To evaluate the accuracy of the dUT1 estimated from Intensive sessions empirically, we require more accurate dUT1 values for comparison at synchronous epochs. One option would be to compare with dUT1 from the IERS 14 C04 series [11]. We will not do this in our study, as dUT1 estimated from Intensive sessions is used as an input to the IERS 14 C04 series. Another option would be to use dUT1 estimated from the simultaneous 24 h VLBI sessions. Although it would be possible to compare dUT1 from 24 h sessions directly, these sessions typically take place 3–4 times a week only, whereas Intensive sessions are carried out every day. Consequently, for this study, we target sessions coming from the continuous VLBI campaign CONT17 (see next sub-section). Moreover, during the CONT17 campaign, the 24 h and Intensive sessions took place on the same day for 15 consecutive days. This provides an excellent opportunity to directly compare dUT1 values from Intensives and legacy 24 h sessions. During this campaign, three kinds of 24 h sessions and three types of Intensive sessions took place. More details regarding EOP estimated during the CONT17 campaign are given in [12].

In this work, we estimated the dUT1 from the individual and combined legacy networks. We kept the temporal resolution to be daily and hourly. In addition, we estimated dUT1 from the different Intensive sessions. We employed different comparison strategies in our study, which will be detailed in Section 2.4. Finally, we also evaluated the dUT1 from VGOS sessions for the five days at daily and sub-daily resolutions.

In Section 2.5, we introduced a novel concept of sub-daily dUT1-tie, intending to improve dUT1 estimation from Intensive sessions and mitigate the sub-daily variations present in dUT1.

### CONT17 Campaign

The CONT17 campaign officially started on 28 November 2017 at 00:00:00 UTC, and it ended on 12 December 2017 at 23:59:59 UTC [13]. The participating stations, along with baselines of the Intensive sessions, can be seen in Figure 1. The Legacy-1 network, also referred to as the IVS network, consists of 14 globally distributed IVS stations. The Legacy-2 network, also referred to as VLBA network, consists of nine stations from the Very Long Baseline Array (VLBA) situated in the USA supplemented by four IVS stations to enhance its global coverage. Besides legacy networks, the campaign contained five days of the VGOS (VLBI Geodetic Observing System) network consisting of six VGOS stations. We excluded the RAEGYEB station since the baselines that include the RAEGYEB station have poor quality (i.e., due to pointing issues at RAEGYEB, the observations were not considered reliable). The information regarding the 24 h sessions can be seen in Table 1. Besides, three types of Intensive sessions took place. The IVS Intensives consisting of IVS-INT1 and IVS-INT2 recorded observations from Wettzell–Kokee baseline. The IVS-INT1 took place at 18:30 UTC and was held during 11 days of the campaign, whereas IVS-INT2 took place at 07:30 UTC and was held on the remaining four days of the campaign. The Russian Intensives recorded observations on the Badary–Zelenchukskaya baseline, two stations that are part of the Russian Quasar Network, and the sessions started after 19:00 UTC. The stations that participated in IVS-Intensives were a part of the Legacy-1 network. They dropped out of the 24 h session to observe the Intensive session and later rejoined the 24 h session. For more details regarding the Intensives, refer to Table 2.

We assigned Legacy-1 as “L1”, Legacy-2 as “L2”, and combined legacy network as “L12”, for convenience. We will use these short names throughout the rest of the article. The number of observations and scans contained in legacy and VGOS networks and different Intensive sessions are illustrated in Figure 2. We can say from Figure 2 that the average number of observations in legacy networks is around 3–4 times that of VGOS. However, the average number of scans between legacy and VGOS sessions are comparable. Although the average number of observations in legacy networks is more than a factor of 1000 as compared with the Intensive sessions (see Figure 2 and Figure 3), the number of scans per hour in the legacy network is only twice the number of scans in Intensive sessions. In addition, the L2 has a higher observation yield with fewer scans, which is an indication that many of the observations are correlated. The statistics do not hold in general but are subject to the scheduling strategy, sub-netting, etc.

## 2. Methodology

The results discussed in this study are derived from VLBI analysis, the details of which will be specified in this chapter. In this work, we used the GFZ version of the Vienna VLBI Software [14], VieVs@GFZ [15], upon which the PORT [16] is based.

For our analysis, we used the latest International Terrestrial Reference Frame 2014 (ITRF2014) [17] and International Celestial Reference Frame 3 (ICRF3) [18]. The rest of the models and the a priori values used for this analysis are compiled in Table 3. While applying for non-tidal atmospheric loading at the observation equation level is the norm in operational VLBI data analysis, it is currently not recommended by the latest IERS Conventions [2]. However, unlike station coordinates, applying or not non-tidal loading models yield differences in the estimated Earth orientation parameters that are not statistically significant [19]. Therefore, we carried out the analysis without correcting for environmental loading effects. We compared the dUT1 values during the campaign with respect to the a priori, i.e., finals.all (IAU2000) solution (ftp://cddis.gsfc.nasa.gov/pub/products/iers/finals2000A.all, accessed on 1 March 2022) and not the posterior values. The dUT1 were estimated as continuous piecewise linear functions (PWLF) in the VieVS@GFZ. For further understanding, we computed the root mean square (RMS) and uncertainty of the formal error values from the Equations (Equation 3) and (Equation 4), respectively.
(3)$$\mathrm{RMS}=\sqrt{\frac{1}{n}\sum_{i=1}^{n}\mathrm{dUT}1_{i}^{2}}$$
(4)$$\sigma_{\sigma }=\frac{1}{\sum_{i=1}^{n}\frac{1}{\sigma_{i}}}$$
where $\mathrm{dUT}1_{i}$ = UT1-UTC for the given epoch and σ = corresponding formal errors of $\mathrm{dUT}1_{i}^{2}$ estimated. We also computed Pearson’s linear correlation coefficient between different dUT1 solutions (see Equation (Equation 5)).
(5)$$r=\frac{\sum \left(x_{i}-\bar{x}\right)\left(y_{i}-\bar{y}\right)}{\sqrt{\sum {\left(x_{i}-\bar{x}\right)}^{2}\sum {\left(y_{i}-\bar{y}\right)}^{2}}}$$
where $x_{i}$ and $y_{i}$ are the dUT1 values from different solutions, and $\bar{x}$ and $\bar{y}$ are the mean values of dUT1 from different solutions.

### 2.1. Assessment of dUT1 from Legacy Networks

The dUT1 estimation from the 24 h sessions follows a two-step procedure: single-session analysis and global solution analysis. In the first step, we analyze various parameters as shown in Table 4 from the individual observing sessions. For the second step, we combine the sessions in the global solution module in VieVs@GFZ (see multi-session analysis column in Table 4), where these sessions are combined on the normal equation level. After this step, we obtain dUT1 time series for 15 days from individual and combined legacy networks. We followed the abovementioned procedure for the estimation of hourly dUT1 values from the legacy networks. However, for estimating hourly dUT1, we fix the CPO to their a priori values due to the correlation between CPO and sub-daily polar motion and dUT1 (ERP).

### 2.2. Assessment of dUT1 from Intensives

In contrast, we do not estimate parameters such as polar motion, CPO, station and source coordinates, and tropospheric gradients when analyzing Intensive sessions because these sessions do not contain enough observations (see Table 4). Besides, the station coordinates cannot be estimated, as one baseline without constraints is insufficient to fix the degree of freedom of the terrestrial basis. Hence, the dUT1 estimates from the Intensive sessions may contain systematics with respect to a solution where all parameters are estimated and also because the modeling carried out at the observation equation level is not perfect. As Intensive sessions contain observations of about one hour only, they give rise to enhanced correlations between CPO and terrestrial pole coordinates. We further grouped the IVS-INT1 and IVS-INT2 as IVS-INT, as these two Intensive types have the same baseline (Kokee–Wettzell). A high relative constraint was applied to dUT1 for stabilizing the dUT1 time series from the Intensive sessions. We assigned short names for IVS Intensives as “IVS-Int” and Russian Intensive sessions as “Ru1-Int” for the sake of convenience.

### 2.3. Assessment of dUT1 from VGOS Sessions

The second part of the study focuses on the assessment of dUT1 from the VGOS sessions with respect to the 24 h legacy sessions. Before analyzing the VGOS sessions, we modified the NGS files (https://cddis.nasa.gov/archive/reports/formats/ngs_card.format, accessed on 1 March 2022) that we created from the associated vgosDB. A VGOS session during the CONT17 campaign spans from 23:00 to 23:00 UTC (next day). We synchronized the start and the end epoch of a VGOS NGS file with respect to a 24 h legacy session, i.e., from midnight to midnight (next day). We removed the first hour of observations, i.e., 23:00 to 00:00 UTC. Then, the observations during 23:00 to 00:00 UTC were added from the next session and so on. However, for the last VGOS session, the last hour of the day remains empty as there is no next session present from which we could add the observations. This is performed to achieve a fair comparison between dUT1 from VGOS and from legacy networks. We follow a similar two-step procedure (see Section 2.1) for estimating dUT1 from the VGOS. Since the VGOS stations are not a part of the ITRF2014, we had to take approximate values for a priori coordinates of these stations (those were taken from the NGS file header) for this analysis. As a result, we observed biases up to 3 mas between the dUT1 obtained from the VGOS network and the dUT1 from the legacy network. This problem should be eliminated in the upcoming ITRF2020, when these VGOS stations obtain coordinates consistent with legacy VLBI stations. Hence, for the comparison, we adjust the mean value of VGOS-derived dUT1 to the mean value of L12-derived dUT1 (during the VGOS days) to maintain the consistency between the legacy and VGOS sessions.

### 2.4. Strategies for Comparison of dUT1 Derived from Different Observing Sessions

We compared and assessed the dUT1 from the Intensive sessions using three different approaches. In the first approach, the dUT1 values from the Intensive sessions were compared with respect to the daily dUT1 from the 24 h legacy networks at the midnight epoch. It is possible that some inconsistencies could be present in the dUT1 from the Intensive sessions due to the different observing times of the Intensive sessions. The estimated dUT1 values from Intensive sessions are computed from observations that span approximately one hour during a day. The dUT1 values at the midnight epoch are extrapolated and could contain significant extrapolation errors.

For the second approach, we compared the dUT1 values derived at the observation interval of the Intensive sessions (synchronized sampling). This approach is based on the estimates of dUT1 from 24 h legacy and Intensive sessions with hourly resolution. Then, we compare the estimated values for the same epoch. For example, if the Intensive session spans from 18:30 to 19:30 UTC, we obtain dUT1 estimates for 18:00, 19:00, and 20:00 integer hour UTC epochs. The determination of three-hourly unconstrained estimates from Intensive sessions is ambitious; however, we would like to try it in this study. Next, we compare it with hourly dUT1 from the legacy network at its mean epoch, i.e., 19:00 UTC. This procedure is applied for both Intensive networks. In this approach, the hourly dUT1 value from the legacy networks, for example, at 19:00 UTC, may be influenced by the neighboring dUT1 values at 18:00 UTC and 20:00 UTC as they are parameterized using piecewise linear offsets in VieVs@GFZ. Therefore, to mitigate this effect, we created pseudo Intensive sessions, which will be discussed in detail in the next paragraph.

In the third approach, we compared dUT1 from Intensive sessions with the artificially created “pseudo Intensive sessions”. We extracted 1 h of observations from the 24 h legacy networks. The start and end epoch of these pseudo Intensive sessions are synchronized with respect to the epoch of the actual Intensive sessions. We analyzed these pseudo Intensive sessions as the standard Intensive sessions (see Table 4). Therefore, for the analysis, we created four sets of pseudo Intensive sessions, i.e., pseudo Intensive sessions from L1 and L2 networks with respect to IVS-Int and Ru1-Int, respectively. We had to create separate pseudo Intensive sessions because of the different epochs of IVS and Russian Intensive sessions (see Table 2). Typically, a pseudo Intensive session consists of around 400–500 observations compared to 30 observations of the actual Intensive session, whereas the pseudo Intensive sessions contain 35–50 scans in comparison to 20–30 scans in the Intensive sessions.

For better understanding, we calculated and compared the RMS differences between dUT1 estimated from the Intensive and 24 h (legacy networks) sessions from the three approaches as discussed above. Using the second and the third approach, we can expect to observe the influence of the piecewise linear parameterization.

### 2.5. Sub-Daily dUT1 Tie

In this section, we introduce a novel approach to improve the dUT1 values estimated from the Intensive sessions. As sub-daily variations are present in dUT1 from Intensive sessions, we aim to minimize these effects. These so-called ties, i.e., correction values (CV), are computed from the 24 h sessions at hourly resolution.

The computation of CV is the difference between the mean values of hourly dUT1 over a day and weighted mean value of dUT1 at the Intensive epochs from 24 h sessions or HFEOP models (Gipson [20] and Desai [21] HFEOP model) (see Equation (Equation 6)). This CV is added to the dUT1 from the Intensives for the corresponding day (see Equation (Equation 7)). This is performed for 15 days of the campaign and separately for Legacy-1 and Legacy-2 sessions.
(6)$${\mathrm{CV}}_{i}={\left(\mathrm{dUT}1_{i}^{avg}-\frac{\mathrm{dUT}1_{i}^{t-1}+2\cdot \mathrm{dUT}1_{i}^{t}+\mathrm{dUT}1_{i}^{t+1}}{4}\right)}_{j}$$
(7)$$\mathrm{dUT}1_{i}^{new}=\mathrm{dUT}1_{i}^{int}+{\mathrm{CV}}_{i}$$
where CV is the correction value, *j* indicates which datasets were used, i.e., 24 h sessions, Desai HFEOP, or Gipson HFEOP, *t* denotes the mean epoch of an Intensive session, and *i* is the number of days (there are 15 days of sessions in the CONT17 campaign). In the case of Intensive sessions, we take weighted mean, i.e., the mean epoch is assigned twice the weight than the preceding and succeeding epoch. For comparison, we computed a reference value, which is the root mean square of the difference between dUT1 from the 24 h session for daily resolution and Intensive session.

## 3. Results and Discussion

### 3.1. dUT1 from 24 h Legacy Sessions

We observed that the dUT1 (daily resolution) from the L2 network have lower uncertainty as compared to the L1 network (see Figure 4 and Table 5). This could be attributed to approximately 1.5 times more observations in the L2 network than in the L1 network. To understand the relationship between dUT1 estimated from L1, L2, and L12 networks, we computed the correlation coefficient between them (see Table 6). For one-day resolution, the correlation between L1 and L2 networks is 0.86. This shows a good agreement between dUT1 estimated from both networks. The correlation coefficient between L2 and combined network (0.98) is higher than in between L1 and L12 networks (0.94). This shows that L2 sessions have slightly larger weight than L1 sessions on the combined network. The dUT1 from the L12 network has significantly smaller formal errors, whereas the variations are about at the same level as L1 (see Figure 5). As shown in Table 5, the RMS of L1 is slightly smaller than that of L12. Note that the RMS of dUT1 estimates with respect to finals.all is not the perfect measure to assess the quality of UT1 estimates. However, since other space geodetic techniques, such as GNSS, do not provide stable UT1, there is no option for external comparison against values of superior quality. Hence, the VLBI-internal assessment as performed in this study is the only possible approach.

The dUT1 (hourly resolution) contains sub-daily variations, which can be seen in Figure 6. The uncertainty of formal errors of hourly dUT1 is significantly higher than the uncertainty of formal errors of daily dUT1 (see Figure 7). This can be due to a smaller number of observations per interval in the case of hourly resolution. For a one-hour resolution, the correlation coefficient between dUT1 from L1 and L2 networks is 0.63 (Table 6). The decrease in correlation for hourly resolution might be due to different amounts of sub-daily variations (i.e., unmodeled signals) in the legacy network. Another possible explanation could be due to the increase in noise caused by parameterization using PWLF.

We also computed dUT1 for various sub-daily resolutions to determine the optimal length for sub-daily dUT1. Considering L1 and L12 solution, 6 h resolution shows highest improvement of around 7% and 2.5% in dUT1 with respect to 1 h dUT1 RMS values, respectively (see Table 7). However, for the L2 solution, 4 h resolution shows the largest improvement of around 5% in dUT1 with respect to 1 h dUT1 RMS values. Upon further analysis, we found that the correlation coefficient considerably improves for 6 h resolution with respect to 1 h for all the cases (see Figure 8).

### 3.2. dUT1 from Intensive Sessions

In this study, dUT1 are estimated from two different Intensive networks: IVS-Int and Ru1-Int. We can observe that dUT1 estimated from Ru1-Int have higher RMS and formal errors as compared to IVS-Int (see Table 5). One of the reasons for this is the different baseline lengths, and dUT1 is sensitive to a more extended east–west baseline. Since Ru1-Int (4364 km in east–west extension) has a much shorter baseline than IVS-Int (10,072 km in east–west extension), this results in higher RMS and formal errors in dUT1.

The dUT1 values estimated from the Intensive sessions along with three legacy solutions are plotted in Figure 5 at midnight epoch. We note that the formal errors are significantly larger from Intensive sessions (see Figure 4). This can be explained as most parameters are not estimated, i.e., fixed to a priori values when estimating dUT1 from Intensives. Hence, any inaccuracies or unmodeled signals present in the a priori values of the fixed parameters will further propagate in the dUT1 determination. The differences in the results (Table 4) are primarily caused by the parameterization of the estimates. The hourly parameterization is more meaningful for the Intensive sessions. To further validate the relationship between dUT1 from Intensive sessions and legacy networks, their correlation coefficient is computed. The correlation coefficient between IVS-Int and the L12 network is 0.04, which indicates no correlation between them. This could be a result of different observing times of the IVS-Int, i.e., 11 sessions at 18:30 and four sessions at 07:30 (see Table 2). The poor correlation between the dUT1 from the different Intensive networks could also be caused by the fact that the Intensive sessions are only observing for one hour, whereas the legacy networks are observing for 24 h.

The second approach was used to compare the hourly dUT1 estimated from the Intensive sessions and 24 h sessions at the mean epoch of an Intensive session. The statistical assessment in Figure 9 reveals important features. The dUT1 from L1, L2, and L12 hourly solutions show an agreement of −2.52%, +7.59%, and +18.60%, respectively, compared to dUT1 from L1, L2, and L12 daily solutions with dUT1 from Intensive sessions. We observe various levels of improvement, and in some cases worsening, for different legacy networks. We repeated the same procedure for dUT1 from the Ru1-Int. However, we observed no improvement when using the second approach for this type of session.

### 3.3. Pseudo Intensive Sessions

In the third approach, we computed the RMS of the differences between dUT1 from the IVS-Int with respect to the pseudo Intensive sessions. We compared these values with the RMS values, which we computed from the first and the second approach and plotted them in Figure 10. We observe that dUT1 from the Intensive sessions show better agreement with pseudo Intensive sessions as compared to the first two approaches. The agreement between the dUT1 from L1, L2 pseudo Intensive, and Intensive sessions is around 13% more than the agreement between dUT1 from 24 h and Intensive sessions. The possible reasons for this better agreement would be the synchronized sampling with the Intensive sessions and the mitigation of effects from preceding and succeeding linear spline segments (due to piecewise linear parameterization).

### 3.4. Sub-Daily dUT1 Tie

We computed new dUT1 values after applying the CV to the dUT1 from Intensives. We estimated the RMS of the differences between the reference value and the three cases. In Figure 11 (left), the CV-added dUT1 showed 13% improvement in agreement with dUT1 from the L1 network as compared to the reference case. Besides, the CV-added dUT1 showed 13% improvement in agreement with dUT1 from the L2 network as compared to the reference case (see Figure 11 (right)). The CV applied to dUT1 that are derived from the HFEOP models shows the highest improvement. The CV (Desai)-added dUT1 showed around 15% and 7% improvement in agreement with dUT1 from the L1 and L2 network as compared to the reference case, respectively. In addition, the CV (Gipson)-added dUT1 showed around 20% and 12% improvement in agreement with dUT1 from the L1 and L2 network as compared to the reference case, respectively.

### 3.5. VGOS

We estimated dUT1 for daily resolution from the VGOS network and assessed its performance against dUT1 from the three legacy solutions plus the combined solutions and Intensive sessions. We can see from Figure 4 that the formal errors from VGOS are comparable to the ones from L1 solution. However, the formal errors from VGOS are approximately two times larger than L2 and L12 solutions. Upon further assessment, dUT1 from VGOS has a correlation coefficient of 0.86, 0.82, and 0.95 with respect to the dUT1 from L1, L2, and L12 solutions, respectively (see Figure 12). From the information above, it is remarkable to mention that the dUT1 from VGOS shows a 95% large correlation with dUT1 from the L12 network. We see such a strong correlation between the VGOS and the legacy networks, even though VGOS sessions contain five stations compared to 28 stations in the combined legacy network. The number of observations in VGOS is less than a factor of nine compared to the combined legacy network. In addition, the number of scans in VGOS is fewer by a factor of two compared to the combined legacy network. This shows basically that VLBI networks with five stations are already capable of producing valuable information at the current level of accuracy and, consequently, a large network of 20–30 stations will considerably improve this kind of EOP determinations. The main reason is the different method of observation, i.e., broadband as opposed to S/X band, in the case of legacy networks. Besides, the VGOS network contains baselines having long east–west extensions, such as Kokee–Wettzell, responsible for dUT1 sensitivity.

In addition, we compared sub-daily dUT1 from VGOS and legacy VLBI. For the hourly resolution of dUT1, we found high formal errors and almost no correlation with respect to the legacy networks. This could be due to the parameterization with linear spline segments (PWLF) for the hourly resolution. In addition, we observed from Figure 7 that the formal errors continuously improve during the VGOS period. The uncertainty of the formal errors on the final day shows an improvement by two times the uncertainty of the formal errors on the first day. We tried for different sub-daily resolutions that would provide us with better results. We computed dUT1 formal errors values from VGOS for various temporal resolutions of integer hours duration. We observed an improvement of 39.83% and 31.61% for 6 and 3 h resolutions in formal errors with respect to 1 h. The 3 h dUT1 from VGOS shows a good correlation of 0.70, 0.65, and 0.75 with L1, L2, and L12, respectively. On the other hand, for 6 h resolution, we observe a strong correlation of 0.78, 0.89, and 0.85 between VGOS and L1, L2, and L12, respectively (see Figure 12).

## 4. Conclusions

We assessed the dUT1 from the CONT17 from various 24 h sessions, such as legacy and VGOS networks, and one-hour Intensive sessions, such as IVS and Quasar VLBI Network (Russian) Intensive networks. The results in this article are an extension and continuation of the previous results contained in [12]. Meanwhile, ref. [12] studied the EOP accuracy when using Legacy-1 network data versus a global solution combining Legacy-1 and Legacy-2 networks; this study provides more in-depth information concerning the assessment of UT1-UTC estimates using different types of sessions (R1, R4, INT), approaches, and parameterization adopted in the VLBI analysis.

Daily-resolved dUT1 from legacy networks shows strong correlation of 0.86. Thus, the dUT1 obtained from the legacy networks is comparable even though the networks have different geometry, stations, scans, and observations. We also evaluated the dUT1 from the legacy networks with hourly resolution. The correlation coefficient between individual legacy networks is 0.62, which is smaller than the correlation of the daily dUT1 values. The possible reason could be due to the involvement of smaller numbers of observations. We observe an increase in the formal errors, and this could be due to the smaller number of observations per parameter, and fewer participating stations. In addition, we assessed dUT1 for various sub-daily resolutions and found that the 6 h resolution performs better than the rest.

Apart from that, we assessed the dUT1 from the two Intensive networks, i.e., IVS (Kokee Park and Wettzell) and Quasar VLBI Network (Badary and Zelenchukskaya) Intensive sessions. The dUT1 estimated from Russian Intensive sessions shows higher RMS and formal errors compared to IVS Intensive sessions. This could be due to the shorter baseline length of Badary and Zelenchukskaya (4364 km) compared to Kokee Park and Wettzell (10,072 km). The dUT1 is sensitive to the baselines with longer east–west extension, and the Kokee Park–Wettzell baseline is about 2.5 times longer than the Badary–Zelenchukskaya.

We further analyzed the three different approaches to compare the dUT1 from the Intensive sessions with the legacy networks. We created pseudo Intensive sessions from the 24 h Legacy-1 and Legacy-2 networks, where start and end epochs were synchronized with the actual IVS and Russian Intensive sessions. For the case of pseudo Intensive sessions synchronized with respect to IVS Intensives, we found an improvement in the agreement of 20.5% and 37.5% for dUT1 from IVS Intensive sessions with respect to the L1 and L2, respectively. We observed a higher amount of agreement compared to the second approach. This is primarily due to the synchronized epoch of the pseudo Intensive sessions. This reveals the shortcomings of parameterization using piecewise linear functions, especially when estimating for sub-daily dUT1.

This work includes the novel idea of implementing sub-daily dUT1-tie to improve the dUT1 from Intensive sessions. The concept involves a CV, which is computed using dUT1 from 24 h sessions (L1 and L2) for hourly resolution. The implementation of the CV derived from Gipson HFEOP sessions improves the highest agreement between dUT1 from IVS-Intensives and L1 sessions by 20% more than the case without CV. In the case of L2 sessions, the improvement in agreement is approximately 12%. The dUT1-tie does seem to improve the estimation of dUT1 from Intensive sessions during the CONT17 period. The time scale for such analysis should be extended to a longer period and is intended for future work.

VGOS sessions took place during the CONT17 period for five days. We assessed the quality of dUT1 from VGOS sessions with respect to the legacy network for these five days. The dUT1 from the VGOS show a high correlation of 0.86, 0.82, and 0.95 with respect to L1, L2, and L12 solutions for daily resolution, respectively. The formal errors of the estimated dUT1 from VGOS are comparable with the L1 solution. The results from the VGOS show strong agreement with the legacy networks despite having fewer observations and only five stations. Therefore, we can say that the VGOS compares well with standard VLBI, even when examining five days of observations only. Hence, it might be challenging to assess the long-term agreement with the legacy networks. In addition, these five sessions are a prototype of limited VGOS with only five stations (the sixth station did not record useful observations), no high-gain schedule (see the small number of scans), and the stations are located in the northern hemisphere only. All this cannot be generalized to judge “real VGOS”. As the VGOS sessions are not explicitly designed for estimating sub-daily dUT1, we computed sub-daily dUT1 from the VGOS for different temporal resolutions. The 6-hourly estimated dUT1 performs better than other resolutions and shows a strong correlation of 0.78, 0.89, and 0.85 with dUT1 from the L1, L2 and L12 6-hourly solutions, respectively.

## Figures and Tables

**Figure 1 sensors-22-02740-f001:**
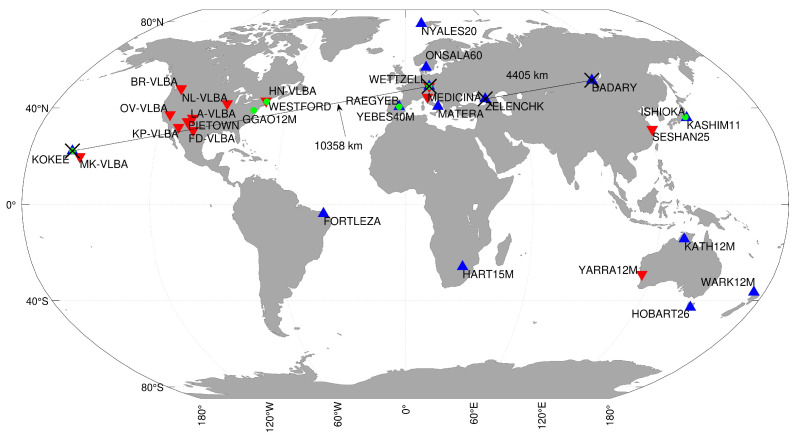
Geographical representation of the stations of the various networks: legacy (S/X) stations in Legacy-1 network stations (upward triangle), Legacy-2 network stations (downward triangle); VGOS stations (green circle); stations that participate in Intensives are indicated by a black cross, and their baselines are represented by a black solid line.

**Figure 2 sensors-22-02740-f002:**
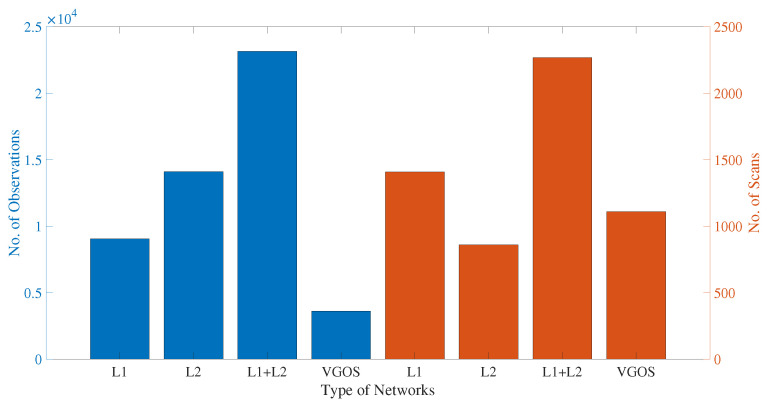
The number of VLBI observations and scans during the CONT17 campaign for 24 h sessions. The *y*-axis (**left**) represents number of observations, and the *y*-axis (**right**) represents number of scans (VGOS number of scans is in between L1 and L2 as it is not yet fully operational).

**Figure 3 sensors-22-02740-f003:**
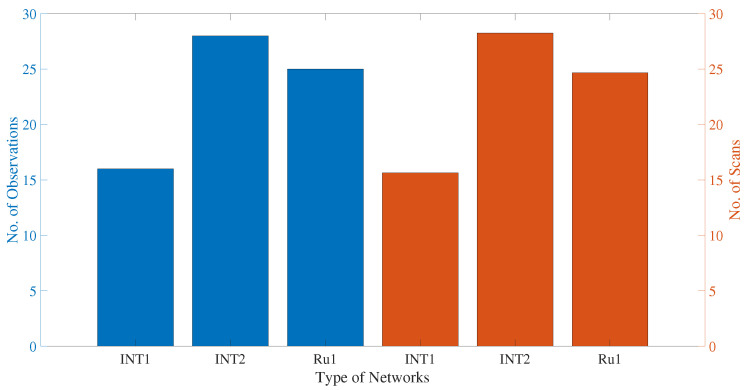
The number of VLBI observations and scans during the CONT17 campaign for Intensive sessions. The *y*-axis (**left**) represents number of observations, and the *y*-axis (**right**) represents number of scans.

**Figure 4 sensors-22-02740-f004:**
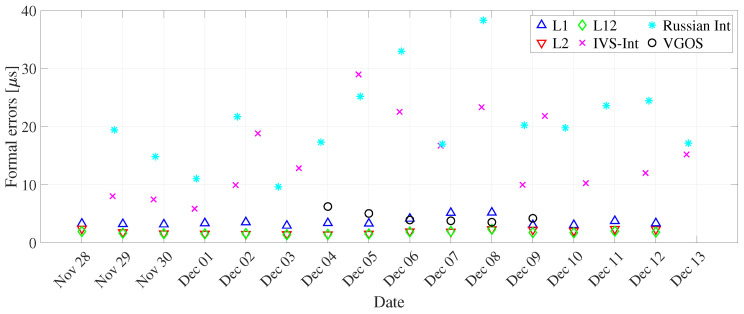
dUT1 formal errors derived from L1, L2, and L12 networks, IVS and Russian Intensives, and VGOS networks for daily resolution (using global solutions).

**Figure 5 sensors-22-02740-f005:**
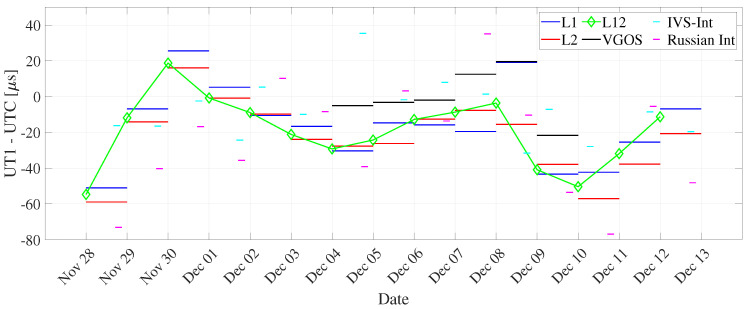
dUT1 estimates with respect to finals.all (IAU2000) derived from L1, L2, and L12 networks, IVS and Russian Intensives, and VGOS networks for daily resolution (using global solutions).

**Figure 6 sensors-22-02740-f006:**
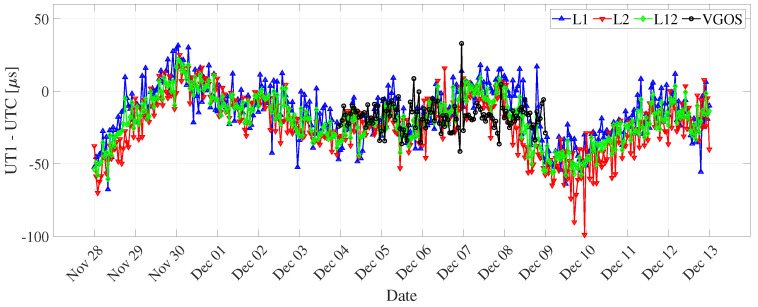
dUT1 estimates with respect to finals.all (IAU2000) derived from L1, L2, and L12 networks, and VGOS networks for hourly resolution (using global solutions).

**Figure 7 sensors-22-02740-f007:**
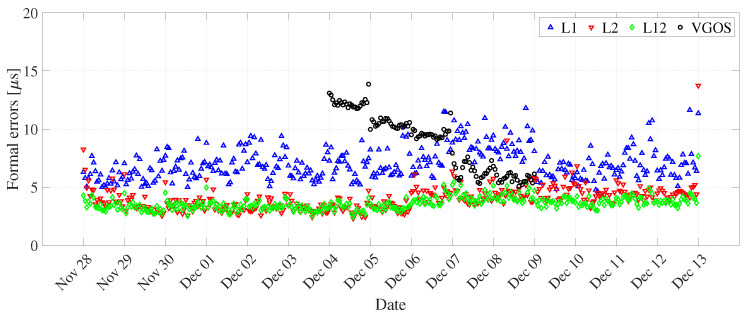
dUT1 formal errors derived from L1, L2, and L12 networks, and VGOS networks for hourly resolution (using global solutions).

**Figure 8 sensors-22-02740-f008:**
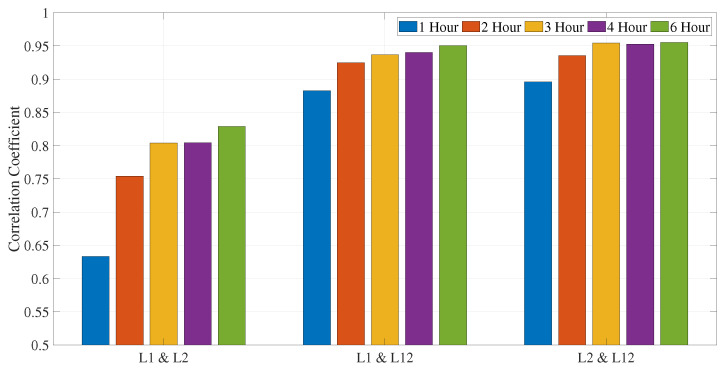
Correlation coefficient of different legacy solutions for different sub-daily resolutions.

**Figure 9 sensors-22-02740-f009:**
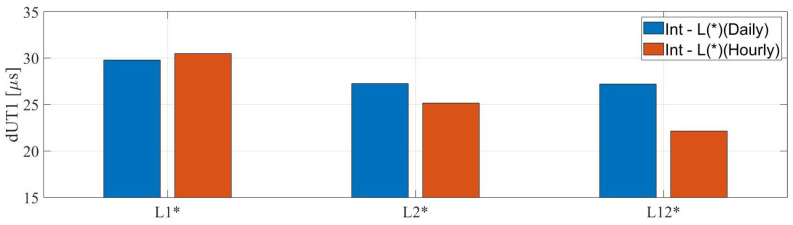
The root mean square values of the differences between daily and hourly dUT1 from the IVS-Int and the 24 h sessions (L1, L2, and L12 networks), represented by blue and red, respectively.

**Figure 10 sensors-22-02740-f010:**
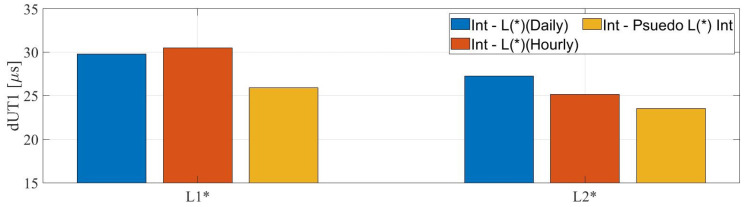
The root mean square values of the differences between daily and hourly dUT1 from the IVS-Int and the 24 h sessions (L1 and L2), represented by blue and red color; the RMS of the differences between dUT1 from IVS-Int and the pseudo Intensive sessions, represented by yellow color.

**Figure 11 sensors-22-02740-f011:**
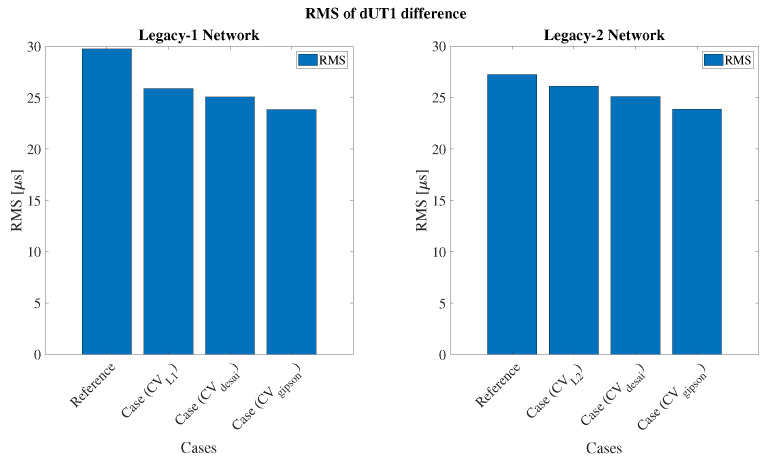
Root mean square of the differences between dUT1 from 24 h and IVS-Intensive sessions (with and without the addition of correction value); ${\mathrm{CV}}_{L1}$, ${\mathrm{CV}}_{L2}$, ${\mathrm{CV}}_{desai}$, and ${\mathrm{CV}}_{gipson}$ are correction values computed with hourly dUT1 from L1 network, L2 network, Desai HFEOP, and Gipson HFEOP, respectively.

**Figure 12 sensors-22-02740-f012:**
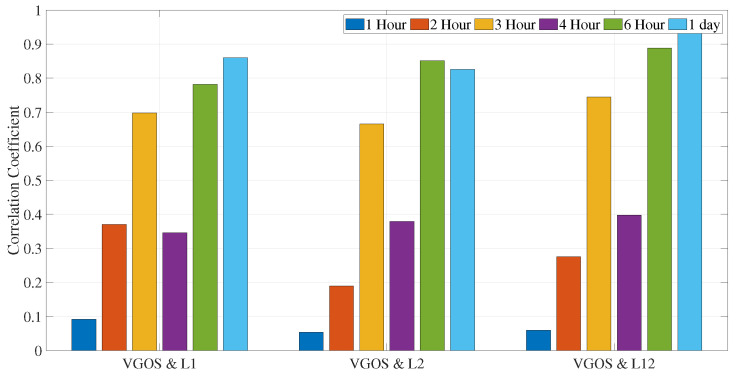
Correlation coefficient of VGOS and legacy solutions for different sub-daily resolutions (analysis performed during five days).

**Table 1 sensors-22-02740-t001:** Overview of 24 h sessions during CONT17.

	Legacy Networks		VGOS Network
	**Legacy-1**	**Legacy-2**	**VGOS**
**Number of Stations**	14	14	6
**Observing days**	28 November–12 December 2017	28 November–12 December 2017	4 December–8 December 2017
**Time frame of observations**	00:00:00 UTC–23:59:59 UTC	00:00:00 UTC–23:59:59 UTC	First 4 sessions: 23:00:00 UTC–22:59:59 UTC (+1), Last session: 23:00:00 UTC–23:59:59 UTC (+1)

**Table 2 sensors-22-02740-t002:** Overview of Intensive sessions during CONT17.

	IVS-INT1	IVS-INT2	Ru1
**Stations**	**Wettzell (Germany) Kokee Park (USA)**	**Wettzell (Germany) Kokee Park (USA)**	**Badary (Russia) Zelenchukskaya (Russia)**
**Length of baseline (km)**	10,357.4	10,357.4	4405
**East-West dimension (km)**	10,072	10,072	4364.7
**North-South dimension (km)**	2414	2414	595.7
**Observing days**	28 November–1 December, 4–8 December, 11–12 December	2, 3, 9, 10 December	28 November–12 December
**Time frame of observations**	18:30–19:30	07:30–08:30	30 November, 4 & 5 December: 19:30–20.30 3 & 11 December: 20:30–21:30 Remaining days: 19:00– 20:00

**Table 3 sensors-22-02740-t003:** Models and a priori series used in the VLBI analysis.

Models/Reference Frame	Type
EOP a priori series	finals.all (IAU2000)
Precession/Nutation Model	finals.all (IAU2000)
Terrestrial RF	ITRF2014
Celestial RF	ICRF3
Ephemerides	JPL421
Pressure	GPT2
Temperature	GPT2
Mapping functions	VMF3
Gradients	APG [9]
Ionosphere delay	From NGS file
Tidal ocean loading	FES2004
Tidal atmosphere loading	vandam
Mean pole model	IERS2015

**Table 4 sensors-22-02740-t004:** Standard parameterization used for analyzing Intensive and 24 h VLBI sessions (constraints are given in brackets, if applicable; N.A. = not applied).

Parameters	Single-Session Analysis (Relative Const., Absolute Const.)		Multi-Session Analysis
	24-h	Intensives	
**PM (x-pole, y-pole)**	Estimated (N.A., N.A.)	Not estimated	Estimated
**UT1-UTC**	Estimated (N.A., N.A.)	Estimated (0.0001 mas, 10 mas)	Estimated
**CPO**	Estimated (N.A., N.A.)	Not estimated	Reduced
**Station Coord.**	Estimated	Not estimated	Reduced
**Source Coord.**	Estimated	Not estimated	Reduced
**Zenith Wet Delay (ZWD)**	Estimated (1.5 cm, N.A.)	Estimated (0.1 cm, N.A.)	Reduced
**Gradients (NGR & EGR)**	Estimated (0.05 cm, N.A.)	Not estimated	Reduced
**Clock**	Estimated (1.3 cm, N.A.)	Estimated (N.A., N.A.)	Reduced

**Table 5 sensors-22-02740-t005:** The root mean square and mean formal error values of dUT1 from legacy, and Intensive sessions for daily and hourly resolution. Unit: μs.

Network		L1	L2	L12	IVS Intensives	Russian Intensives
	**Resolution**					
RMS [μs]	1 day	26.09	29.50	27.15	18.00	39.05
	1 h	23.45	30.17	25.45		
Formal errors [μs]	1 day	5.20	2.42	2.30	28.99	38.31
	1 h	11.80	13.73	7.67		

**Table 6 sensors-22-02740-t006:** Correlation coefficient of the dUT1 estimates from legacy networks for daily and hourly resolution.

Temporal Resolution	Daily			Hourly		
Networks	L1 and L2	L1 and L12	L2 and L12	L1 and L2	L1 and L12	L2 and L12
Correlation Coefficient	0.86	0.94	0.98	0.63	0.88	0.90

**Table 7 sensors-22-02740-t007:** Performance of dUT1 RMS values for 2, 3, 4, and 6 h resolutions with respect to 1 h resolution (increase in [%]). The 1 h values are provided in terms of their RMS values (μs), and the other sub-daily resolutions correspond to the improvement in percentages.

	L1	L2	L12
**1 h**	23.44	30.17	25.45
**2 h**	4.55%	3.13%	1.80%
**3 h**	5.79%	4.44%	1.76%
**4 h**	6.09%	4.77%	2.04%
**6 h**	6.91%	4.56%	2.50%

## Data Availability

Data is available upon request to correspondence author.

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
