# Peer review of "Inter-Comparison of UT1-UTC from 24-Hour, Intensives, and VGOS Sessions during CONT17"

_sensors, 2022, doi:10.3390/s22072740_

Round 1

Reviewer 1 Report

Strings 109-111 “For convenience, we assigned legacy-1 as…” should be placed before the Figures 2,3 references.

Strings 145-153. The first strategy is not clear. The sub-daily variations can be removed from dUT1 values according the IERS conventional procedure (Chapter 8) from both hour and 24-hour dUT1 values. These values refer to the middle (as example) of hour or other interval and valid up to a few tenths of a microsecond in UT1. Why this method can not be used? So the sentence “Note that the sub-daily variations …” (the strings 152-153) does not true. And main problem is to interpolate the dUT1 values to the midnight epoch with microsecond accuracy. It is clear from strings 228-240 and fig.5.

For the second approach the authors calculate “dUT1 estimates for 18:00, 19:00…” (strings 157-159). It is not clear how can get dUT1 estimates for 18:00 and 20:00? They are extrapolated values that contain the extrapolation errors. And is it necessary to do if the authors use only 19:00 dUT1 estimate later?

Result (strings 241-248) can be explained by parametrization of unknowns (table 4). Station coordinates for Intensive sessions are not estimated, but correction in baseline length of order of 1-2 cm leads to a tenths of  microseconds in UT1.

A novel approach to improve the dUT1 values (strings 182-189) is some kind of numerical integration of function. It is not clear what is advantage of this method specially for periodic function such as dUT1? These periodic variations can be removed before integration, and you get almost linear function.

Author Response

Point 1: Strings 109-111 “For convenience, we assigned legacy-1 as…” should be placed before the Figures 2,3 references.                                                     Response 1: Thank you for the comment. We added this statement before figures 2,3 references, i.e., beginning of the second paragraph in section 1.1 (see lines 86-87). We will use these short names throughout the rest of the article.”.

Point 2: Strings 145-153. The first strategy is not clear. The sub-daily variations can be removed from dUT1 values according the IERS conventional procedure (Chapter 8) from both hour and 24-hour dUT1 values. These values refer to the middle (as example) of hour or other interval and valid up to a few tenths of a microsecond in UT1. Why this method can not be used? So the sentence “Note that the sub-daily variations …” (the strings 152-153) does not true. And main problem is to interpolate the dUT1 values to the midnight epoch with microsecond accuracy. It is clear from strings 228-240 and fig.5.              Response 2: Thank you for raising an important point. We are aware of this, but we intended to quantify this effect. We have removed the sentence “Consequently, the sub-daily variations in dUT1 are present in them. Note that the sub-daily variations in daily dUT1 from legacy networks are not present as these variations are mostly averaged out in 24-hour sessions”, as it might mislead the reader. Besides, we added the following sentence “The dUT1 values at the midnight epoch are extrapolated and could contain significant extrapolation errors” to make the readers aware (see lines 155-156).

Point 3: For the second approach the authors calculate “dUT1 estimates for 18:00, 19:00…” (strings 157-159). It is not clear how can get dUT1 estimates for 18:00 and 20:00? They are extrapolated values that contain the extrapolation errors. And is it necessary to do if the authors use only 19:00 dUT1 estimate later?                                                                                                                Response 3: Thanks, that's indeed a good question. dUT1 estimates for 18:00 and 20:00 were modeled as continuous piecewise linear offsets with no constraints. We are aware that the computation of unconstrained dUT1 from Intensive sessions is ambitious, but we wanted to undertake it anyway. To give readers more background, we added the following sentence “ The determination of three-hourly unconstrained estimates from Intensive sessions is ambitious; however, we would like to try it in this study” (see lines 162-163).

Point 4: Result (strings 241-248) can be explained by parametrization of unknowns (table 4). Station coordinates for Intensive sessions are not estimated, but correction in baseline length of order of 1-2 cm leads to a tenths of  microseconds in UT1.                                                                                      Response 4: Yes, the reviewer is right that this can be explained by the parametrization of unknowns. To improve understanding for the reader, the following sentence was added to section 3.2: “The differences in the results (Table 4) are primarily caused by the parameterization of the estimates. The hourly parameterization is more meaningful for the intensive sessions” (see lines 241-243)

Point 5: A novel approach to improve the dUT1 values (strings 182-189) is some kind of numerical integration of function. It is not clear what is advantage of this method specially for periodic function such as dUT1? These periodic variations can be removed before integration, and you get almost linear function.       Response 5: We appreciate the reviewer’s comment, however, we do not observe it as a "numerical integration" here. We used sub-daily EOP models / observed high-frequency values to compute the difference between Intensive sessions and 24h sessions. This difference is then applied to correct the sub-daily dUT1 from the Intensive session, further improving the comparison between 24h and Intensive results. The advantage of this method is that sub-daily variation is a periodic function that changes with time. With our approach, we compute the difference and remove it. This is what we call a "sub-daily dUT1 tie". The values from Intensive sessions corrected in this way we can, for example, be combined with the values from 24h sessions thereafter.

Reviewer 2 Report

Figure 1: I would suggest to match it with Table 1. Legacy (S/X) stations in VLBA network (downward triangle), IVS network (upward triangle) --- Legacy-1 network stations (upward triangle), Legacy-2 network stations (downward triangle)

Equation 3 and 4: As the whole paper is focused on UT1 but not include polar motion, these two equations concerning ERP appear irrelevance, not mentioned in follow up discussions. Anyway, RMS is used in many cases but not defined clearly.

Line 130: It is not clear how to modify NGS files. Just to remove the first hour of data?

Line 137: Here the a priori coordinates of VGOS stations were taken from the NGS file header, resulting in 3 mas bias in UT1 estimates, approximately 100 times bigger than UT1 formal errors. I would suggest to adopt better positions of VGOS stations deduced from global solutions, or add a reference paper to show that large positional errors made little impact on the estimation of UT1, in terms of UT1 formal errors and its time variation.

Line 201: It is difficult to judge that the dUT1 from the L12 network has less variations from Figure 5. As listed in Table 5, the RMS value of L12 is bigger than that of L1. Moreover, the RMS (dUT1 estimates w.r.t. finals.all) here is not suitable to assess the quality of UT1 estimates for L1, L2 and L12.

Line 215: The dUT1 RMS value should be defined or explained here or somewhere, e.g. in section 2.

Line 246: As the dUT1 from L1, L2, and L12 hourly solutions shows an agreement of -2.52%, +7.59% and +18.60% respectively, you can not say there are various levels of improvement (both positive and negative values are presented) in the case of different legacy networks.

Author Response

Point 1: Figure 1: I would suggest to match it with Table 1. Legacy (S/X) stations in VLBA network (downward triangle), IVS network (upward triangle) --- Legacy-1 network stations (upward triangle), Legacy-2 network stations (downward triangle)                                                                                    Response 1: Thank you for pointing it out. We have modified the caption of Figure 1 with Legacy-1 network stations (upward triangle), Legacy-2 network stations (downward triangle).

Point 2: Equation 3 and 4: As the whole paper is focused on UT1 but not include polar motion, these two equations concerning ERP appear irrelevance, not mentioned in follow up discussions. Anyway, RMS is used in many cases but not defined clearly.                                                                                                 Response 2: We have acknowledged the point raised by the reviewer and replaced ERP with dUT1 in equation 3 to prevent any confusion.

Point 3: Line 130: It is not clear how to modify NGS files. Just to remove the first hour of data?                                                                                                Response 3: We appreciate identifying a part missing from the article. We have added the following sentence “We removed the first hour of observations, i.e., 23:00 to 00:00 UTC. Then the observations during 23:00 to 00:00 UTC were added from the next session and so on. However, for the last VGOS session, the last hour of the day remains empty as there is no next session present from which we could add the observations”. (see lines 133-137)

Point 4: Line 137: Here the a priori coordinates of VGOS stations were taken from the NGS file header, resulting in 3 mas bias in UT1 estimates, approximately 100 times bigger than UT1 formal errors. I would suggest to adopt better positions of VGOS stations deduced from global solutions, or add a reference paper to show that large positional errors made little impact on the estimation of UT1, in terms of UT1 formal errors and its time variation.        Response 4: We acknowledge that the reviewer is correct. We are aware of this, and accordingly, we removed the bias in UT1 estimates. As the a priori coordinates are the same for each session, we assume that the effect on dUT1 is just a constant offset. To remove this bias, significantly better a priori positions would be of great help. For our study, where we just compare the dUT1 relatively, it is not necessary to do such extensive work beforehand from our point of view.

Point 5: Line 201: It is difficult to judge that the dUT1 from the L12 network has less variations from Figure 5. As listed in Table 5, the RMS value of L12 is bigger than that of L1. Moreover, the RMS (dUT1 estimates w.r.t. finals.all) here is not suitable to assess the quality of UT1 estimates for L1, L2 and L12.             Response 5: Thank you for your comment. We do agree that the RMS of dUT1 estimates w.r.t. finals.all is not optimal for measuring the quality of dUT1 estimates, but we know that there is a lack of an external comparison of dUT1, therefore, VLBI-internal assessment is the only possible method. We modified the following sentence “The dUT1 from the L12 network has smaller formal errors and less variations (see figure 5)" to "The dUT1 from the L12 network has significantly smaller formal errors, whereas the variations are about at the same level of L1 (see figure 5). As shown in table 5, the RMS of L1 is slightly smaller than that of L12. Note that the RMS of dUT1 estimates w.r.t. finals.all is not the perfect measure to assess the quality of UT1 estimates. However, since other space geodetic techniques, such as GNSS, do not provide stable UT1, there is no option for external comparison against values of superior quality. Hence, the VLBI-internal assessment as performed in this study is the only possible approach" (see lines 205-211)

Point 6: Line 215: The dUT1 RMS value should be defined or explained here or somewhere, e.g. in section 2.                                                                               Response 6: Thank you for the comment. As you can see in the article that we have already mentioned regarding the RMS (also see equation 4). We believe that the provided information should be sufficient for the scope of the article.

Point 7: Line 246: As the dUT1 from L1, L2, and L12 hourly solutions shows an agreement of -2.52%, +7.59% and +18.60% respectively, you can not say there are various levels of improvement (both positive and negative values are presented) in the case of different legacy networks.                                    Response 7: The reviewer is right that this was expressed less clearly. Consequently, it can be misleading to the readers. Therefore, we have modified the sentence to “We observe various levels of improvement, and in some cases worsening for different legacy networks” from “We observe various levels of improvement in the case of different legacy networks” (see lines 255-257)